# Mixed Design Optimization of Polymer-Modified Asphalt Mixtures (PMAs) Containing Carton Plastic Packaging Wastes

Riccardo Monticelli, Antonio Roberto *, Elena Romeo and Gabriele Tebaldi *

Department of Engineering and Architecture, University of Parma, Parco Area delle Scienze, 181/a,
43121 Parma, Italy; riccardo.monticelli@unipr.it (R.M.); elena.romeo@unipr.it (E.R.)
* Correspondence: antonio.roberto@unipr.it (A.R.); gabriele.tebaldi@unipr.it (G.T.)

**Abstract:** This study investigated the potential of reusing plastics derived from multilayered food carton recycling processes as a modifier for asphalt mixtures by dry process. Two types of plastics, one untreated and one pelletized, were added to a mixture containing neat asphalt binder using three different dry methods and in two different amounts (2 and 5% by weight of aggregates). The chemical and physical properties of the plastics were explored via differential scanning calorimetry (DSC) and laser diffraction granulometry. The fracture behavior of the polymer-modified asphalt (PMA) mixtures was evaluated via superpave indirect tensile (IDT) testing at 10 °C and comparative analysis with the reference and a mixture composed of the same aggregate curve and a traditional styrene–butadiene–styrene (SBS)-modified asphalt binder. The role of the plastic during crack initiation and propagation was investigated via microscopic analysis. The results show that plastics derived from carton recycling processes can be successfully used as an additive in asphalt mixtures via the dry method due to the compatibility between their melting and mixing temperatures. The shape of the plastics influences the cracking propagation and resistance of the mixture. Finally, the presence of plastics in the mixture reduces the proneness to accumulate deformation (about 50% less than the reference ones) and increases the failure resistance, leading to a better cracking response at intermediate temperatures.

**Keywords:** polymer-modified asphalt mixtures; plastics; food packaging; superpave IDT; cracking response

## 1. Introduction

Environmental sustainability is a major concern in road infrastructure, which significantly contributes to global warming due to the large amounts of energy consumed for their construction, the use of raw materials, and the generation of a large amount of waste. In recent decades, many efforts have been devoted to increasing the use of waste materials in pavement construction while not compromising the qualities of the pavements themselves [1–10]. Both the U.S. government and the European Commission promote the use of cleaner materials and sustainable technologies to extend the life cycle of pavement. For these reasons, the market has been driven to explore new products that provide resistance and durability with technically feasible solutions while minimizing the impact on the environment. The most widely used recycling material in the field of road pavement is recycled asphalt pavement (RAP) for the production of new asphalt mixtures. However, recently, the concept of recycling has been moving toward a focus on reusing waste materials [1] otherwise destined for landfill, such as waste glass [11,12], steel slags [13,14], tires [15–17], propylene waste [18,19], and waste polyester [20,21]. One of the waste materials involved in the recycling process of all economic fields is plastic, which is nowaday one of the most widespread pollutants in the world. Annual plastic consumption is growing daily: around 26 million tons of plastic waste are generated in Europe every year, but less than 30% of such waste is collected for recycling [22]. Plastic for food packaging accounts

for more than one-third of production [23]. A good solution for waste plastics reuse is their introduction as modifiers in asphalt pavement materials [24]. The use of some virgin resins as asphalt binder modifiers (i.e., poly (ethylene-co-vinyl acetate) EVA and poly(styrene-co-butadiene-co-styrene [SBS])) has already been established for several years for producing better-performing asphalt pavement [25]. In recent decades, recycled thermoplastic resins, such as polyethylene (PE), polypropylene (PP), and polyethylene terephthalate (PET), have also been evaluated as modifiers because they offer high mechanical and thermal properties, being also very cost-effective [26]. This leads to the assumption that waste plastics can be successfully reused in asphalt mixtures; however, to date,a lot of uncertainty still persists about the correct amount and the correct introduction during mixing procedures [26,27]. It must also be taken into account that recycled plastics show high variability in their properties compared with virgin plastics, especially in rheological and thermal aspects [28]. Many physical properties, such as density, viscosity, and melting point, may vary depending on the recycling processes or on the presence of residues from the separation of plastics in multilayered packaging [29,30]. It is therefore of crucial importance to correctly select the type of plastics to be added to asphalt mixtures and to identify the most appropriate methodology for introducing that particular type of plastic into the mixture. The National Center for Asphalt Technology (NCAT), along with a task force from the National Asphalt Pavement Association (NAPA) and the Asphalt Institute (AI), conducted a comprehensive literature review on the use of recycled plastic in asphalt materials in 2019 [31,32]. This study was then recently updated as part of the National Cooperative Highway Research Program (NCHRP) project 9-66 [33]. The review contains all the results obtained on the use of plastics in asphalt materials, highlighting knowledge and gaps in knowledge. According to the NCAT report, plastics can be incorporated into asphalt mixtures via a wet process, i.e., mixed with the asphalt binder, or via a dry process, as aggregates (this can be achieved in different ways). Plastics with a low melting point, such as PE and PP, are incorporated via the wet method. In this case, low-viscosity plastics should be preferred because they provide better dispersion in the asphalt binder [27]. It should be emphasized that, with the wet method, it is not recommended to incorporate more than 2% of plastic by weight of asphalt binder to avoid possible phase separation [34,35]. Conversely, high-melting-point and amorphous plastics, such as PET, are added into asphalt mixtures via the dry method, i.e., inserted during mixing, similar to aggregates. Due to the viscoelastic properties of PET particles, the methodology of inserting the plastics into the mixture is crucial to ensure proper dispersion and does not affect the workability or compaction of the final mixture, especially when PET is used at high percentages (>4%) [24,27]. Plastics inserted via the dry process can play different roles within the mix depending on its physical properties and the mixing methodology. They can act as simple substitutes for the aggregates or partially as modifiers of the asphalt binder and aggregate coating, without, however, dissolving completely in the asphalt binder as occurs with the wet process. In this case, parts of the plastics act as a quasicontinuous phase in the mastic, and others become a discontinuous part of the skeletal lithic phase [31,32,36]. The size and shape of the PETs can also influence the mechanical and volumetric properties of the final asphalt mixture [37]. Several studies have shown that the addition of recycled plastics via the dry process makes the asphalt mixture stiffer, with increased resistance to rutting [38–41], but the effects on fracture resistance and on the rate of damage accumulation at the typical intermediate temperatures at which fatigue phenomena start and evolve have not yet been investigated.

## 2. Objectives and Scope

The main objective of this study was to investigate the possible reuse of plastics derived from multilayered food carton recycling processes as a modifier for PMA asphalt mixtures added via the dry process. The focus was on assessing the method of plastic incorporation in order to obtain a mixture with good volumetric and workability properties and better performance than traditional polymer-modified mixtures. In particular, the effects of the presence of plastics on the fracture behavior and rate of damage accumulation

of asphalt mixtures in the viscoelastic field at intermediate temperatures were evaluated. The role of the plastic during crack initiation and propagation was investigated via mycroscopic analysis. Two types of plastics from the food packaging recycling process, one untreated and one pelletized, were inserted into a mixture containing a natural asphalt binder by employing three different types of dry process. The chemical and physical properties of the plastics were explored via differential scanning calorimetry (DSC) and laser diffraction granulometry. The fracture behavior of the PMA mixtures was evaluated using the superpave indirect tensile (IDT) testing at 10 °C.

## 3. Materials and Methods

Food packaging cartons are composed of several layers, each one made of different materials, such as plastic, paper, and aluminum foil. Typically, this type of waste is recycled through a maceration process, which allows the separation of the different layers, enabling both aluminum and paper to be recovered and recycled. At the end of this process, the recovered plastic is coarsely extruded and featured by both densified polymeric grains and plastic fibers, which are typically destined for incinerators or other recycling supply chains. This material is mainly characterized by a grain size of 0.0–5.0 mm and a density of 1.00–1.05 g/cm$^3$. The chemical and physical properties of the plastics were explored via differential scanning calorimetry (DSC) and laser diffraction granulometry.

### 3.1. Chemical and Physical Properties of Plastics

The chemical analysis of the material was performed analyzing the plastics' thermal changes using DSC. DSC is a thermal analysis technique used to study the influence of temperature variations on the heat capacity ($C_p$) of a material. Changes in the heat capacities of a sample of known mass, when heated or cooled, are plotted as changes in heat flux. This allows the highlighting of transitions such as melting, glass transitions, or phase changes. DSC measurements were obtained using a DSC 600 Perkin Elmer instrument in the 10–450 °C range, at 10 °C$^{-1}$ both in nitrogen and air atmosphere. Particle size analyses were performed with a Mastersizer 3000 laser diffraction granulometer (Malvern Instruments®, Malvern, UK), which can effectively evaluate particles with equivalent diameters from 10 nm up to 3500 µm. Each sample was analyzed with both air (Aero S) and water (Hydro EV) dispersion units to best characterize the particle size distribution in the different dispersion media. Optical granulometers exploit the diffraction of light produced by laser radiation, which intersects the particle flow within the analysis cell. The incident laser beam is produced by two different light sources with different characteristic wavelengths. Red light is generated by a He-Ne source, with a current power of 4 mW and a wavelength of 632.8 nm. Blue light is generated by a LED source, with a power of 10 W and a wavelength of 470 nm. Optical diffraction was carried out by adopting Mie diffraction theory [42], which requires the refractive and absorption indices of the analyzed materials.

### 3.2. Asphalt Mixtures' Preparation and Plastic Incorporation

The mixtures were prepared via the dry method, resulting in PMA mixtures. The materials involved were a 50/70 neat asphalt binder (N), a 3.5% SBS polymer modified asphalt binder (M), limestone filler, and two types of food packaging plastics: raw (RP) or pelletized (PP). The used plastics were supplied from a carton food packaging recycling process. Figure 1 shows the two different plastics employed, which were mainly composed of low-density polyethylene (LDPE), high-density polyethylene (HDPE), and polypropylene (PP). The asphalt binders were characterized via their performance grade (PG), resulting in 58-22 and 64-22 for the neat and modified binders, respectively. The plastics were used in two different amounts: 2 and 5% (by weight of aggregates), separately, evaluating their effects on the performance level of the mixtures in comparison with that of the reference ones. These materials were mechanically mixed to obtain 152 mm cylindrical specimens characterized by an aggregate gradation (4500 g virgin aggregates) with a maxi-

mum nominal aggregate size (NMAS) of 12.5, an asphalt binder binder content of 5.2%, and a filler–binder ratio of 1.5. Figure 2 shows the grain distribution of the aggregates used.

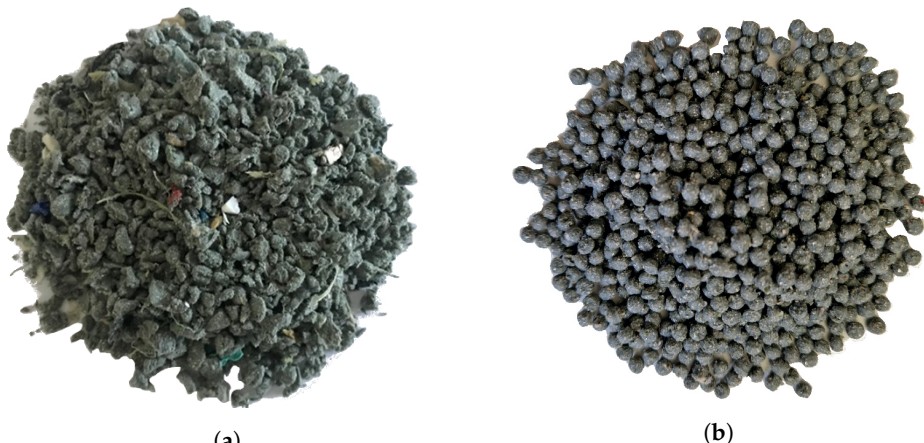

(**a**)  (**b**)

**Figure 1.** Raw (**a**) and pelletized (**b**) plastics.

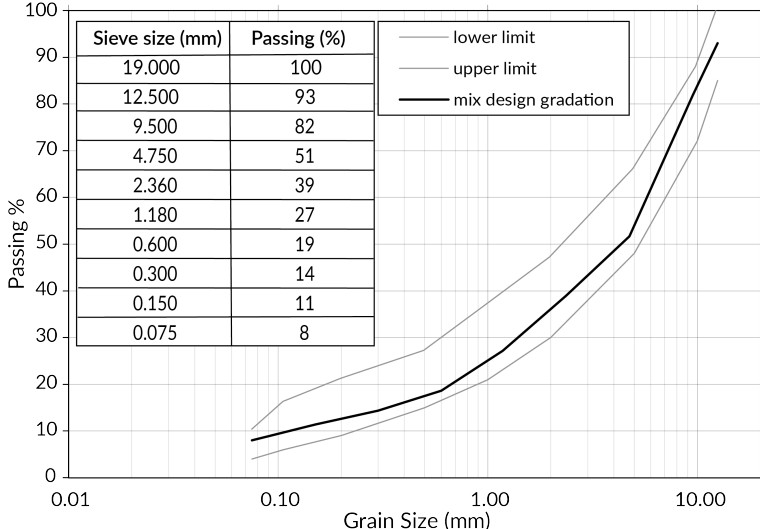

| Sieve size (mm) | Passing (%) |
|---|---|
| 19.000 | 100 |
| 12.500 | 93 |
| 9.500 | 82 |
| 4.750 | 51 |
| 2.360 | 39 |
| 1.180 | 27 |
| 0.600 | 19 |
| 0.300 | 14 |
| 0.150 | 11 |
| 0.075 | 8 |

**Figure 2.** Grading curve of prepared HMAs.

For both dosages of plastics, three different mixing methods were investigated, varying the procedure used for plastic addition and paying attention to the possible development of fumes due to overheating of the polymeric material. In the first method, plastics were added cold to preheated virgin aggregates. Next, asphalt binder was inserted; and, finally, filler was added. The second method featured the insertion of plastics after mixing the preheated aggregates and asphalt binder, with a final addition of filler. The third mixing method consisted of adding plastics to virgin aggregates before the preheating phase for 4 h at 170 °C. After this time, the aggregate plastics were mixed with asphalt binder and then with filler. The nomenclature for the mixing methods is as follows:

M1:  cold plastics added after aggregates;
M2:  cold plastics added after asphalt binder;
M3:  plastics heated together with aggregates for 4 h at 170 °C.

The PMAs were then placed in a dry oven at 135 °C for 2 h, simulating the short-term aging process. The mixtures were finally compacted using a gyratory compactor by imposing 126 revolutions, resulting in air void content varying from 3 and 7%. The mixtures were then allowed to cool for 12 h before being cut to obtain circular-shaped specimens with a thickness of 35 mm. The material combinations and mixture labels are summarized in Table 1.

**Table 1.** Summary of the material combinations and mixture labels.

| Label | Asphalt Binder | Aggregates and Filler | Type of Plastic | Plastic Content | Plastics' Mixing Procedure |
|---|---|---|---|---|---|
| NL | Neat | Limestone | − | − | − |
| ML | 3.5% SBS polymer modified | Limestone | − | − | − |
| RP1_2% | Neat | Limestone | RP | 2% | M1 |
| RP1_5% | Neat | Limestone | RP | 5% | M1 |
| RP2_2% | Neat | Limestone | RP | 2% | M2 |
| RP2_5% | Neat | Limestone | RP | 5% | M2 |
| RP3_2% | Neat | Limestone | RP | 2% | M3 |
| RP3_5% | Neat | Limestone | RP | 5% | M3 |
| PP1_2% | Neat | Limestone | PP | 2% | M1 |
| PP1_5% | Neat | Limestone | PP | 5% | M1 |
| PP2_2% | Neat | Limestone | PP | 2% | M2 |
| PP2_5% | Neat | Limestone | PP | 5% | M2 |
| PP3_2% | Neat | Limestone | PP | 2% | M3 |
| PP3_5% | Neat | Limestone | PP | 5% | M3 |

NL-ML: reference mixtures containing neat (N) asphalt binder and polymer-modified (M) asphalt binder. RP-PP: mixtures containing raw plastic (RP) and pelletized plastic (PP). M1-M2-M3: plastics' mixing procedure.

### 3.3. SuperPave IDT Test

To investigate the cracking behavior of the PMA mixtures in the visco-elastic regime, the superpave indirect tensile (IDT) test was performed at 10 °C [43–45]. This protocol allows for the evaluation of a material's elastic response, the proneness to accumulate permanent deformations, and the resistance to cracking and failure at intermediate temperatures. Tests of resilient modulus [45], creep-compliance [44], and tensile strength [43] were sequentially performed. In accordance with the HMA Fracture Mechanics Framework [46–48], it is possible to estimate the work required to initiate a nonhealable fracture, namely, the fracture energy (FE) and its components, defined as elastic energy (EE) and dissipated creep strain energy (DCSE), respectively, at the point of first fracture.

## 4. Results

This section discusses the results of both chemical–physical properties and cracking behavior.

### 4.1. DSC Analysis

The results of the DSC analysis are reported in Figure 3. It is possible to see that the dry process started for the analyzed plastics (at 82.28 °C). The DSC curve could be basically characterized by four peaks, associated to the solid–liquid phase transition. Particularly, LDPE, HDPE, and PP are characterized by low melting points, 104.05 °C, 124.36 °C, and 159.73 °C, respectively. The other two peaks are typically related to PET melting. Thus, the compatibility between the mixing temperatures of the plastics and PMAs was assessed, even though the mixing temperature did not allow for a fully melted condition of the plastics. Therefore, they partially acted as aggregates in the PMAs.

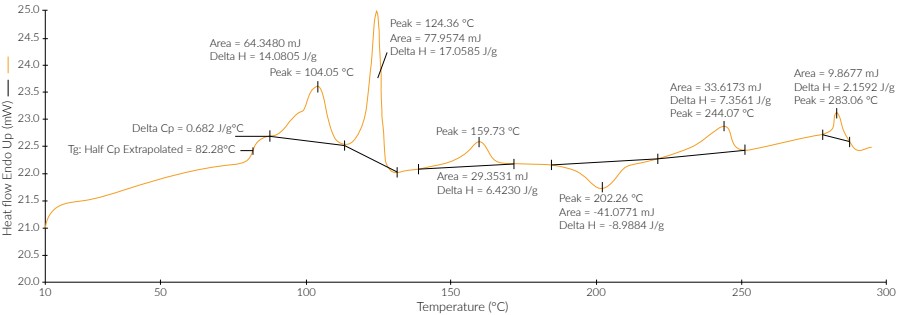

**Figure 3.** Results of DSC analysis of plastics.

## 4.2. Particle Size Distribution

The results of particle size analysis did not show significant differences between the pelletized (PP) and raw (R) plastics. Their similar grain distribution is shown in Figure 4. As expected, the specific surface area (SSA) was much higher for pelletized plastics: 3.898 $m^2$/kg (R) and 4.253 $m^2$/kg (PP). Figure 4 also shows that R was characterized by small particles between 100 and 500 μm, indicating that nanoparticles could be found in the sand fraction of the aggregates.

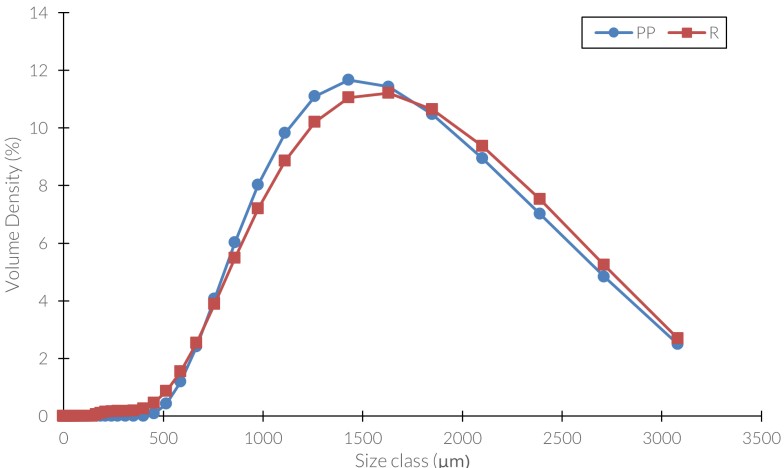

**Figure 4.** Particle size distribution of raw and pelletized plastics.

## 4.3. Workability of PMAs

The compactability and workability of the PMA mixtures were examined during compaction with a gyratory compactor. The density curves are shown in Figure 5. As can be observed, the combination with the lowest workability was PP2_2%, i.e., the mixture containing 2% pelletized plastics and mixed via method 2. This was most likely due to the low adhesion between aggregates and plastic grains. In general, workability was reduced when plastics are added to mixtures, either pelletized or raw, for both percentages used (2 and 5%). However, at $N_{des}$ (N = 126), the density curves of the mixtures without plastics (NL and ML) were comparable to the PP3_2%, PP1_2%, and PP2_5% mixes. Looking at the magnification of the graph (Figure 5), it can be seen that the RP1, RP2, and RP3 mixtures containing 2 or 5% plastic were able to meet the void percentage required by the mix design of 6 ± 0.5%. In general, the introduction of plastics worsened the workability of the mix, but the raw plastics added via method 1 still allowed the mix to maintain good compactability and workability.

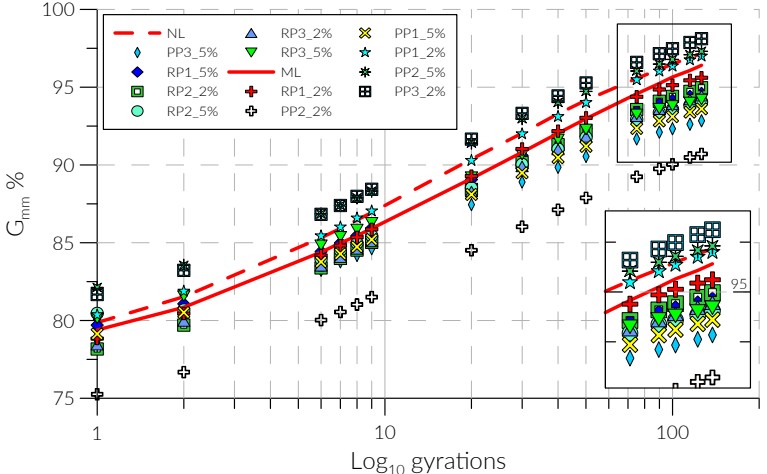

**Figure 5.** Analysis of PMAs' workability using a gyratory compactor.

### 4.4. Microscopy Analysis

Microscopic analysis was performed to investigate the plastic shape or failure mode of the LDCs. Therefore, for each tested specimen, three sections of the cracking edge were prepared by dividing the tested specimens. Then, images were taken at different zoom scales, as shown in Figure 6. As expected, the results showed the different behaviors of the plastics both during the mixing process and after cracking. Indeed, for blends M1 and M2, the plastics formed a kind of veil in the aggregate skeleton of PMAs for both raw and pelletized plastics (up to 2%). During the cracking process, elongated plastic fibers could be detected. This phenomenon was enhanced for the R1_2% blend, as shown in Figure 6. Conversely, the M3 blend showed a different plastic failure, which was rather brittle when compared with the previous one. It seems that it did not soften the plastics enough, not allowing the formation of plastic ribbon. Because of this, the cracking edge of the plastic did not show elongated fibers, indicating a poor level of deformation before cracking. Microscopic analysis also showed aluminum particles embedded in both the raw and pelletized plastics.

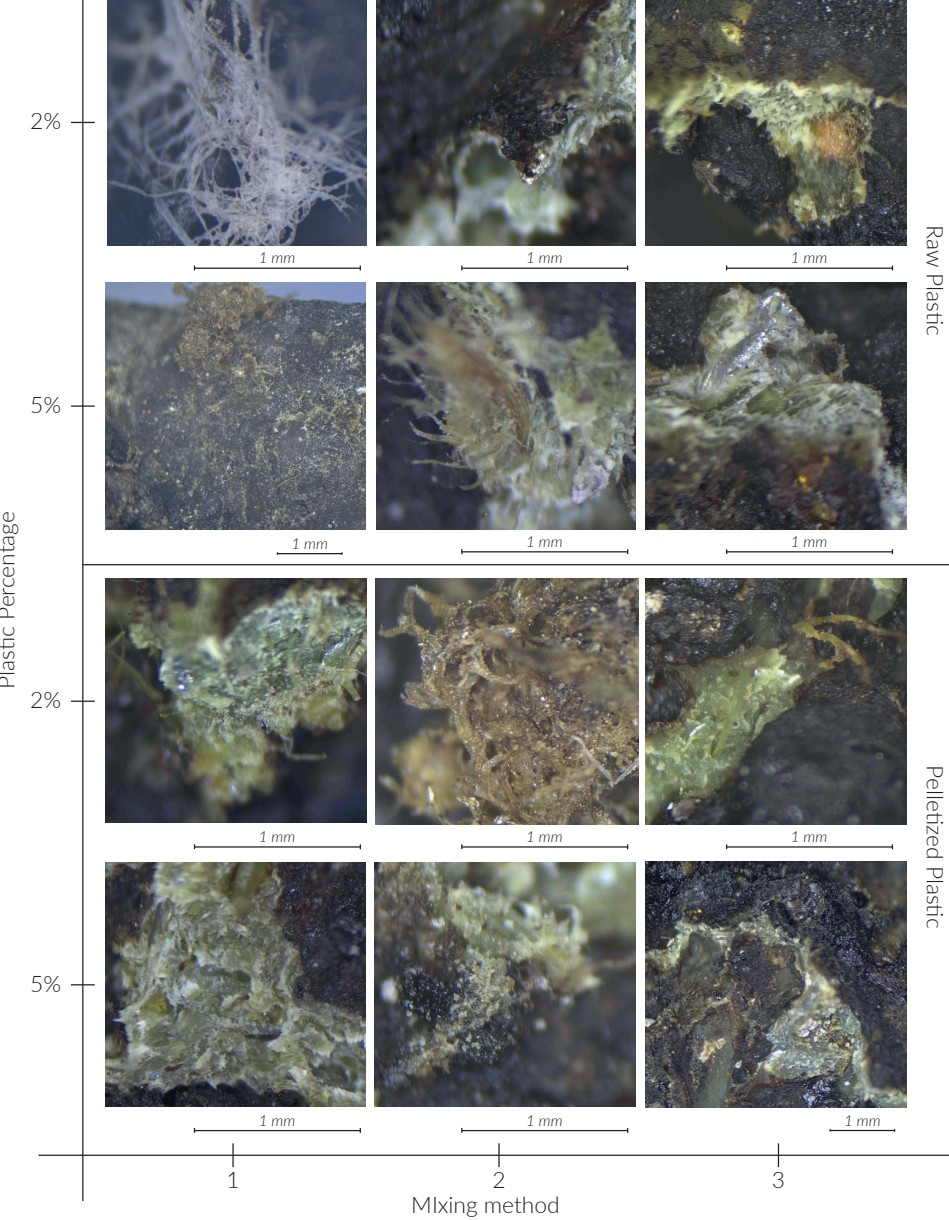

**Figure 6.** Results of the microscopy analysis performed on the cracking edge of the tested specimens.

### 4.5. Superpave IDT—Cracking Response

The analysis of the PMA's performance level involved three tests, which were sequentially performed at 10 °C: resilient modulus, creep compliance, and tensile strength. These allowed us to investigate the elastic response of the investigated materials, their proneness to accumulate permanent deformation, and their failure resistance.

### Resilient Modulus

The results of the resilient modulus test are summarized in Figure 7. Comparing the mixtures without plastics with the the others, no significant differences were found between them if the plastic content did not exceed 2% for either raw or pelletized plastics and or among the mixing methods. Mixtures containing 5% raw plastics were quite similar to the others, indicating no significant differences in the mixtures' lithic skeleton. Conversely, mixtures containing 5% pelletized plastic showed an elastic response that was strongly dependent on the mixing method adopted.

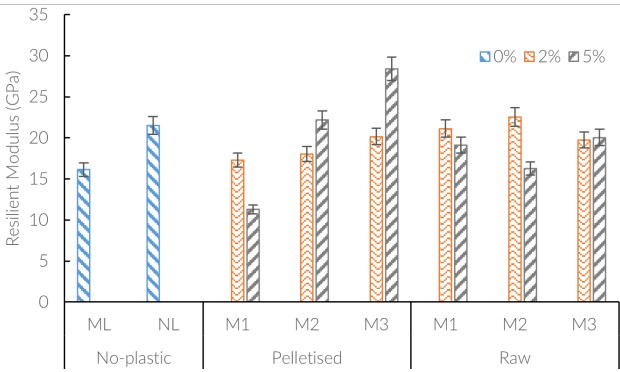

**Figure 7.** Results of the Resilient Modulus test at 10 °C.

### 4.6. Creep Compliance

The results of the creep compliance test are reported in Figure 8. As expected, the proneness of accumulate permanent deformation was reduced by up to 50% when either pelletized or raw plastic was added to the mixture (Figure 8). Lower creep compliance values were obtained in mixtures to which we had added raw plastics: the higher the percentage of plastic, the lower the strain accumulation. Moreover, the mixtures with pelletized plastics showed quite similar behavior at both 2 or 5% contents. Despite this, the mixtures prepared using M1, containing 2% plastics, had a higher creep compliance than the others. The m value, which represents the creep rate of the material, did not differ among the mixtures containing 2% plastics and the no-plastic mixtures for each mixing method, indicating that the deformation accumulated at the same rate. Conversely, the m value exponentially increased with a 5% pelletized plastic content. This finding may be associated with the higher plastic content in the skeleton of the mixture.. The raw plastics basically had a lower m value for both plastic contents, and each mixing method produced a value matching that of the reference SBS asphalt mixture (ML).

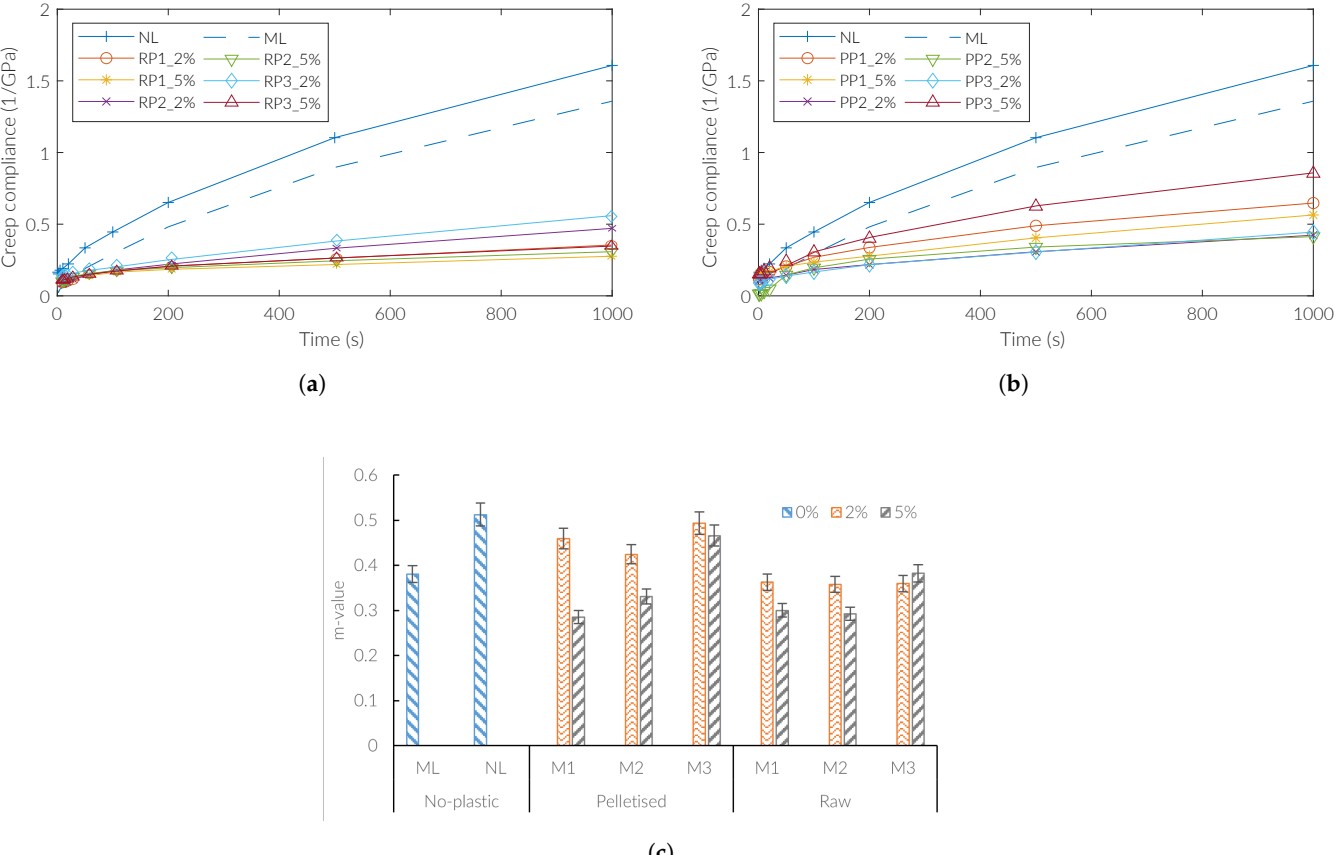

**(a)**

**(b)**

**(c)**

**Figure 8.** Results of the creep compliance test at 10 °C, showing the creep compliance curves (**a**,**b**) and m-value (**c**) parameter.

### 4.7. Tensile Strength

The results of the tensile strength test are reported in Figure 9. Mixtures containing 2% of pelletized plastics showed no significant differences in fracture resistance for the adopted mixing procedures. Conversely, at the higher plastic content (5%), significant differences were found among the mixing methods, particularly between M1 and M3. Comparing mixtures with added pelletized plastics with the NL reference mixture, a decrease in tensile strength was observed. On the other hand, when comparing the results of mixtures admixed with raw plastics, no significant differences were noted between methods M2 and M3 when the plastic content was 5%, while method M1 led to an important decrease in tensile strength. In contrast, the results observed for mixtures with 2% raw plastic were quite similar to those observed for the reference mixtures. The tensile strength of the SBS-modified mix (ML) was lower than that of the others; this could indicate a tendency for the plastics to stiffen the mixtures. However, this hypothesis was not supported by the failure strain results, which showed that at the point of failure, corresponding to the fracture initiation point, the additivated mixes showed a higher deformation than the virgin reference. The most deformable mixtures were those containing 5% raw plastic and prepared via method M1. Comparing the results with those of the ML-modified SBS mixture, the only additive mixture that had both higher tensile strength and failure strain was the one containing 5% raw plastic obtained via method M1. However, it is important to emphasize that the plastic-additivated mixtures achieved values very close to those of the SBS-modified one. Figure 10 shows energy threshold results. In terms of FE, the raw plastics were able to improve the required energy to fracture the mixture with a single load application, particularly when the mixtures were prepared via M1. The energetic analysis also highlighted the important effects of the plastics on the DCSE, indicating that the plastics reduced the accumulation of permanent deformation. Moreover, the EE was improved when plastics were used in the mixture for all mixing methods. Nevertheless, for the 5% pelletized M1 mixture, the EE was below that of the ML.

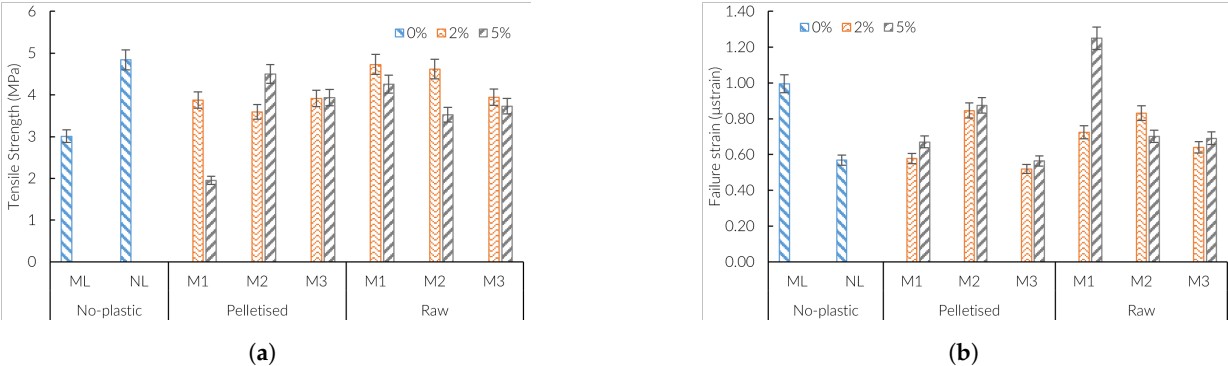

**Figure 9.** Results of the tensile strength test at 10 °C, showing tensile strength (**a**) and failure strain (**b**) parameters.

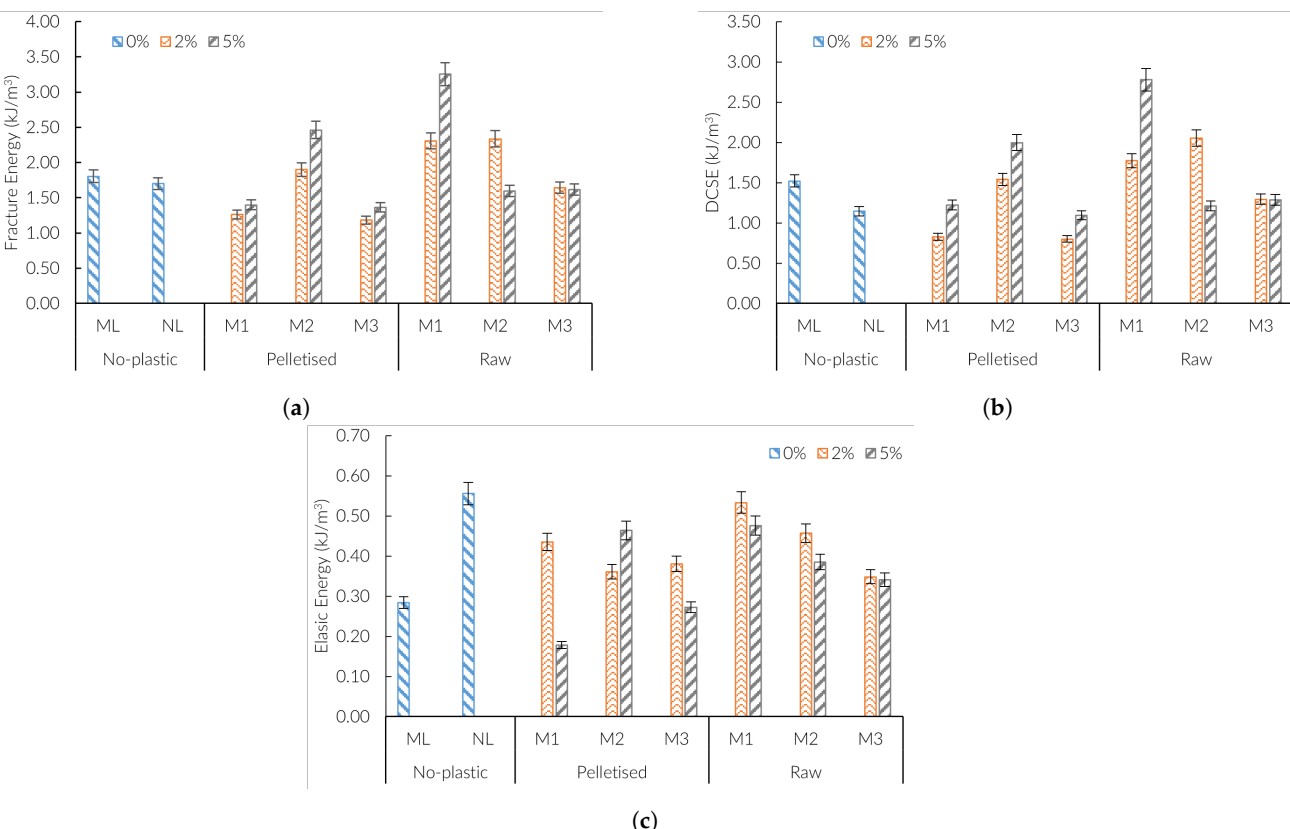

**Figure 10.** Results of the energetic analysis, showing fracture energy (**a**), DCSE (**b**), and elastic energy (**c**) of the analyzed materials.

## 5. Summary and Conclusions

The present study aimed at evaluating the effects of the additivation to mixtures containing natural asphalt binder od plastics derived from the food packaging recycling process. Pelletized and raw plastics were used at 2 and 5% by weight of the total aggregates. These were compared with both virgin and SBS-modified reference mixtures. Three mixing methods were explored to evaluate the role this addition can play in the final performance of the PMA mixture: method 1—placing the plastics together with the hot aggregates; method 2—adding the plastics after the asphalt binder; and method 3—preheating the plastics to mixing temperature before adding them to hot aggregates. DSC analysis was performed to assess the cross-matching between the plastic's melting point and the mixing temperatures generally used for the dry method. Superpave IDT testing was performed at intermediate temperature to evaluate the cracking response of the additivated mixtures in terms of elasticity, proneness to accumulating deformation, tensile resistance, and energy thresholds.

The edge of cracking sections was also analyzed using a high-resolution microscope to investigate the role of the different plastics in cracking initiation and propagation. The outcomes can be summarized as follows:

- DSC showed compatibility between the mixing temperature of the mixtures and the melting temperature of the analyzed plastics, indicating the possibility of their reuse as asphalt modifier using the dry method.
- The microscopy analysis highlighted different shapes and cracking modes of the plastics. Particularly, while the raw plastics exhibited stretched fibers along the plastics' cracking edge, the pelletized ones showed quite brittle failure, especially when the mixture was prepared using M .
- Superpave IDT analysis showed that mixtures containing plastics, either pelletized or raw, were generally characterized by a low proneness to accumulating permanent deformation (about 50% less than the reference ones), despite of that fact the creep rate was quite similar to that of the analyzed standard mixture. Considering the mixtures containing plastics, this was better highlighted at the failure point, where they were generally characterized by a high deformation before mixture failure (about 20% considering RP1_5%).

It can be concluded that plastics from the food packaging carton recycling process can be used as an additive for asphalt mixtures containing neat asphalt binder. The cracking response of PMA mixtures containing the plastics was comparable to that of an SBS-modified mixture with an average modification content. In general, it was observed that both the shape of the plastics and the method of inserting them into the mixture, as well as the amount of plastic added, play a significant role in the final performance. It was concluded that pelletized plastics, inserted into a cold mix after hot aggregates, at a high percentage by weight of the aggregates (5%), yield better values both in terms of resistance to the accumulation of permanent damage and deformability before fracture initiation in the viscoelastic field. Such a mixing mode, as the microscopic pictures suggest, allows the plastics to act as substitutes for the aggregates and as modifiers of asphalt binder. Further improvements will involve in-scale analysis of traffic effects on these mixtures, with a focus on rutting and its mitigation with plastic addition. Furthermore, the microstructure of LDCs will be analyzed.

**Author Contributions:** Conceptualization, A.R., R.M., E.R. and G.T.; methodology, A.R., R.M., E.R. and G.T.; formal analysis, A.R., R.M., E.R. and G.T.; data curation, A.R. and R.M.; writing—original draft preparation, A.R. and E.R.; writing—review, A.R., R.M., E.R. and G.T.; editing, A.R.; project administration, A.R., R.M., E.R. and G.T. All authors have read and agreed to the published version of the manuscript.

**Funding:** Financial support from PNRR MUR project ECS_00000033_ECOSISTER is acknowledged.

**Acknowledgments:** The authors acknowledge the technicians for their technical support.

**Conflicts of Interest:** The authors declare no conflict of interest.

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
