# Peer review of "Mixed Design Optimization of Polymer-Modified Asphalt Mixtures (PMAs) Containing Carton Plastic Packaging Wastes"

_sustainability, doi:10.3390/su151310574_

Round 1
Reviewer 1 Report
Manuscript
Journal Name: Sustainability, MDPI
Journal Version: Sustainability, 2023, 1, 0
Title of manuscript: Mix-design optimization of polymers-modified asphalt mixtures (PMAs) containing carton-plastic-packaging wastes
Type of manuscript: Research article
The following query and advice should be considered.
1- It should be noted where the materials are supplied. (plastic, binder, aggregate and etc.)
2- Where the material support is received and where the experimental work is conducted?
3- Have any mixing processes been carried out for materials?
4- Some long sentences within the article should be simplified to better understand.
5- The materials should be explained for the purpose and short technical specifications.
6- English spelling and grammatical corrections should be reviewed.
7- Abbreviations should be shown under Table 1.
8- What was considered in the dry and wet process?
9- The standards used should be given.
10- Repeated sentences should be avoided.
11- A flow chart should be given.
12- The physical properties of aggregate and bitumen should be presented as a table.
13- The properties of the plastic material should be presented as a table.
14- The first paragraph in the summary and conclusion sections should be transferred to the result section.
15- The explanation of Figure 2 should be clearly stated again.
16- Page 8, lines 243-244 statement should be revised and edited about creep compliance test result.
17- Figure 2 should be revised and given with threshold values.
18- What does it mean? (condiering, Page 11 line 302)
19- The authors should clarify the use of the SBS-modified asphalt they mentioned.
20- What is the granulometry of raw and pelletized plastic materials? The dimensions used should be specified.
21- Aggregate gradation should be given with a graph.
22- Function equations of Figure 2 should be given.
23- What kind of future studies should be done?
24- Why was the choice of plastic content taken as 2% and 5% while preparing the mixture?
25- Why was only short-term aging applied in the preparation of the material? Why is long-term aging not done to express real conditions?
26- How much of an economic contribution does the use of plasticine as recycling have compared to normal mixtures?
As a result of the above evaluations, it appears that the manuscript is understandable in terms of its structure, and fiction but it is necessary that the above recommendation should be considered. Because of that, the review opinion of the manuscript should be improved and reassessed by making some revised. For this reason, this article is need to major correction at this stage.
Manuscript
Journal Name: Sustainability, MDPI
Journal Version: Sustainability, 2023, 1, 0
Title of manuscript: Mix-design optimization of polymers-modified asphalt mixtures (PMAs) containing carton-plastic-packaging wastes
Type of manuscript: Research article
The following query and advice should be considered.
1- It should be noted where the materials are supplied. (plastic, binder, aggregate and etc.)
2- Where the material support is received and where the experimental work is conducted?
3- Have any mixing processes been carried out for materials?
4- Some long sentences within the article should be simplified to better understand.
5- The materials should be explained for the purpose and short technical specifications.
6- English spelling and grammatical corrections should be reviewed.
7- Abbreviations should be shown under Table 1.
8- What was considered in the dry and wet process?
9- The standards used should be given.
10- Repeated sentences should be avoided.
11- A flow chart should be given.
12- The physical properties of aggregate and bitumen should be presented as a table.
13- The properties of the plastic material should be presented as a table.
14- The first paragraph in the summary and conclusion sections should be transferred to the result section.
15- The explanation of Figure 2 should be clearly stated again.
16- Page 8, lines 243-244 statement should be revised and edited about creep compliance test result.
17- Figure 2 should be revised and given with threshold values.
18- What does it mean? (condiering, Page 11 line 302)
19- The authors should clarify the use of the SBS-modified asphalt they mentioned.
20- What is the granulometry of raw and pelletized plastic materials? The dimensions used should be specified.
21- Aggregate gradation should be given with a graph.
22- Function equations of Figure 2 should be given.
23- What kind of future studies should be done?
24- Why was the choice of plastic content taken as 2% and 5% while preparing the mixture?
25- Why was only short-term aging applied in the preparation of the material? Why is long-term aging not done to express real conditions?
26- How much of an economic contribution does the use of plasticine as recycling have compared to normal mixtures?
As a result of the above evaluations, it appears that the manuscript is understandable in terms of its structure, and fiction but it is necessary that the above recommendation should be considered. Because of that, the review opinion of the manuscript should be improved and reassessed by making some revised. For this reason, this article is need to major correction at this stage.
Author Response
- It should be noted where the materials are supplied. (plastic, binder, aggregate and etc.)
Answer: Thank you very much for your kind suggestion, we indicated in the text this information.
- Where the material support is received and where the experimental work is conducted?
Answer: Dear we have written the main information related to materials and methods. We thought that where the experimental work is conducted and where the material support is received are not essential information. Thus, it should not influence the quality of the manuscript. Anyway, as you can recognise in the authors’ affiliation, the study proposed is conducted at The University of Parma, Department of Engineering and Architecture, Laboratory of asphalt pavement infrastructures, structures, and environment.
- Have any mixing processes been carried out for materials?
Answer: Thank you very much for this question. As you can recognise along the text, the main topic of this manuscript is the analysis of the correct mixing process to produce PMAs. It is well specified in the text at the objective and scope section. Particularly, PMAs have been compared to HMAs in terms of performance levels to assess whether PMAs can behave similarly to HMAs or not.
- Some long sentences within the article should be simplified to better understand.
Answer: Thanks for the suggestion. The article has been revised following this observation.
- The materials should be explained for the purpose and short technical specifications.
Answer: Thanks for your clarification request. Materials’ specifications are reported in the materials and methods section, where each component has been described. In order to give as much information as possible about technical specifications the asphalt binder PGs have been reported.
- English spelling and grammatical corrections should be reviewed.
Answer: Thanks for the suggestion. The article has been revised following this observation.
- Abbreviations should be shown under Table 1.
Answer: Thanks for the suggestion. The article has been revised following this observation.
- What was considered in the dry and wet process?
Answer: The differences was stated in the introduction. What we reported is : “According to the NCAT report, plastics can be incorporated into asphalt mixtures by a wet process, i.e. mixed with the asphalt binder, or by a dry process, as aggregates (this can be done in different ways)”.
- The standards used should be given.
Answer: Thanks for highlighting this aspect. As you can recognize in the SuperPave IDT test section, we reported the references of the three tests. Those are the ones on which the SuperPave protocol bases on.
- Repeated sentences should be avoided.
Answer: Thanks for the suggestion. The text has been revised.
- A flow chart should be given.
Answer: Thanks for this suggestion. Because of we described the project in the objective and scope section, we avoided to add a project flow chart, limiting the use of pictures to materials, charts, and graphs.
- The physical properties of aggregate and bitumen should be presented as a table.
Answer: Thank you for your request. The physical properties of aggregate and bitumen are described within the text, trying to be as descriptive as possible. As per the previous comment, we used images to show the main material, plastics in this case, and its properties, gradation, and morphology.
- The properties of the plastic material should be presented as a table.
Answer: Thank you for your suggestion. As per the previous comment, we emphasised the properties, such as particles distribution, which can affect HMA behaviour.
- The first paragraph in the summary and conclusion sections should be transferred to the result section.
Answer: Thank you for this tip. As I understood correctly, you requested to move the study’s summary in the results section. This is not clear. We thought that the results are not related the study’s summary.
15- The explanation of Figure 2 should be clearly stated again.
Answer: Thanks for the suggestion. The article has been revised following this observation.
16- Page 8, lines 243-244 statement should be revised and edited about creep compliance test result.
Answer: Thanks for the suggestion. The article has been revised following this observation.
17- Figure 2 should be revised and given with threshold values.
Answer: Thank you very much for this question. We do not understand what thresholds you are referring to. The results showed are all referred to the same material with the same characteristic and composition. Thus, the DSC results are perfectly overlapped.
18- What does it mean? (condiering, Page 11 line 302)
Answer: Thanks. We corrected the typo, it is considering.
19- The authors should clarify the use of the SBS-modified asphalt they mentioned.
Answer: Thank you for your observation. SBS- modified asphalt binder was used to create the reference mixture. The reason for this mixture is to compare the modifying effect of plastics with natural bitumen. The results indicated that the plastics act as a modifier of natural bitumen.
20- What is the granulometry of raw and pelletized plastic materials? The dimensions used should be specified.
Answer: Dear as you can recognize along the text, granulometry is associated to the laser diffraction test we performed. Thus, it is the test used to establish the grain distribution of the materials you mentioned in your comment.
21- Aggregate gradation should be given with a graph.
Answer: Thank you for your advice. We added the required grading curve.
22- Function equations of Figure 2 should be given.
Answer: Thank you for your comment. Since the reported chart is referred to direct measurements we are not able to provide equations for the drawn curve.
23- What kind of future studies should be done?
Answer: Thank you for the valuable question. Currently, we are in the process of testing the behaviour of mixtures at high temperatures. At the same time, we are setting up a test field to test the mixture under the effects of traffic. In addition, thanks to the cooperation of the plastics recycling company, we are creating a business plan to support this reuse. We added the description of the further improvements in the text.
24- Why was the choice of plastic content taken as 2% and 5% while preparing the mixture?
Answer: These two percentages were used because previous studies in the literature have shown that the contribution of plastics maximizes when used within this range.
25- Why was only short-term aging applied in the preparation of the material? Why is long-term aging not done to express real conditions?
Answer: Thank you, that is a very good point. We only performed short-term ageing before compacting because it is mandatory required by the SuperPave protocol.
26- How much of an economic contribution does the use of plasticine as recycling have compared to normal mixtures?
Answer: Thank you for the valuable question. Currently, we are in the process of testing the behaviour of mixtures at high temperatures. At the same time, we are setting up a test field to test the mixture under the effect of vehicular traffic. In addition, thanks to the cooperation of the plastics recycling company, we are creating a business plan to support this reuse.
Reviewer 2 Report
The authors investigated the potentiality of re-using plastics derived from multilayered food carton recycling processes as a modifier for asphalt mixtures by dry process. This research achievement contributes to the treatment of packaging waste and has certain environmental significance. However, there are still a few questions that readers need to express clearly。
(1) Packaging plastics are generally made of polyolefin materials, such as polyethylene and polypropylene. Can these materials be well mixed with asphalt and have compatibility?
(2) Different plastics have different molecular weights and their DSC curves. How does the author think about this issue?
(3) In this paper, too much consideration has been given to engineering mixing problems, and the essential problems of materials science have not been involved, such as molecular structure, molecular compatibility, interface structure, etc. The research results are somewhat superficial.
(4) Relevant data needs to be provided regarding the microstructure after mixing.
The language meets the requirement of this Journal.
Author Response
(1) Packaging plastics are generally made of polyolefin materials, such as polyethylene and polypropylene. Can these materials be well mixed with asphalt and have compatibility?
Answer: Thank you for your question. Yes, previous studies have shown that there is compatibility. In addition, the plastic introduced with Dry methodology has different roles within the asphalt mix. These aspects were highlighted in the paper.
(2) Different plastics have different molecular weights and their DSC curves. How does the author think about this issue?
Answer: Thank you for the interesting question. We have tried to make this explicit in the 'introduction' paragraph.
(3) In this paper, too much consideration has been given to engineering mixing problems, and the essential problems of materials science have not been involved, such as molecular structure, molecular compatibility, interface structure, etc. The research results are somewhat superficial.
Answer: Thank you for the comment. We apologize about that. Nevertheless, as you can recognize in our affiliation, we are part of the Department of Engineering and Architecture, Thus, the study proposed aims at engineering the reuse of waste plastic materials through the optimization of the mixing process. Therefore, we focussed on analysing the match between the mixing temperature of PMAs and softening point of waste plastics. Moreover, the chemical affinity between asphalt binder and plastic is actually assessed in literature due to their hydrocarbon nature.
(4) Relevant data needs to be provided regarding the microstructure after mixing.
Answer: Thank you for this tip. We added this point in the description of further improvements.
Reviewer 3 Report
Dear authors
Please consider following major comments:
1-Please add important numerical results in abstract and conclusion sections.
2-Please write full name of SBS in abstract section.
3-In line 38, you used resin phrase for introducing two important modifiers for asphalts. The mentioned materials are two kinds of rubbery copolymers. Their correct complete names are poly(ethylene-co-vinyl acetate) and poly(styrene-co-butadiene-co-styrene). Please correct them in text.
2-In materials and methods section, please explain about kind of food packaging cartons that you used in this study. For example, tetrapacked packaging. Please add the response in abstract and conclusion sections in addition to experimental section.
2-Please explain about kind of polymer which was recycled from food packaging cartons. For example, polyethylene or cellulose and so on. Please add the response in abstract and conclusion sections in addition to experimental section.
3- In DSC section, you said results obtained from recycled polymers were belonged to three various polymers i.e., PE, PP, and PET. Did you use these recycled polymers separately or a mixture of them? If you used a mixture of them, did you investigate results of using separate recycled polymers and reasons of differences in obtained results?
4-Please add some original test curves obtained from various samples analysis (creep, tensile, and so on) as in DSC analysis. You just showed final results in charts.
5-There are numerous spelling and grammatical errors that must be corrected.
Sincerely
There are numerous spelling and grammatical errors that must be corrected.
Sincerely
Author Response
1-Please add important numerical results in abstract and conclusion sections.
Answer: Dear the discussion of results was done along the results section. Therefore, summary and conclusion section is used to summarize the study approach and then point-out the main outcomes. The abstract must be limited to comment the results in a descriptive way.
2-Please write full name of SBS in abstract section.
Answer: Thanks for the suggestion. The article has been revised following this observation.
3-In line 38, you used resin phrase for introducing two important modifiers for asphalts. The mentioned materials are two kinds of rubbery copolymers. Their correct complete names are poly (ethylene-co-vinyl acetate) and poly(styrene-co-butadiene-co-styrene). Please correct them in text.
Answer: Thanks for the suggestion. The article has been revised following this observation.
4- In materials and methods section, please explain about kind of food packaging cartons that you used in this study. For example, tetrapacked packaging. Please add the response in abstract and conclusion sections in addition to experimental section.
Answer: Thanks for the suggestion. Within the paper was a description of the composition of the plastic and where it came from. Unfortunately, due to copyright restriction at this stage, it was not possible to proceed further.
5- Please explain about kind of polymer which was recycled from food packaging cartons. For example, polyethylene or cellulose and so on. Please add the response in abstract and conclusion sections in addition to experimental section.
Answer: Thanks for this request. The chemical composition of the plastics was reported along the text.
6- In DSC section, you said results obtained from recycled polymers were belonged to three various polymers i.e., PE, PP, and PET. Did you use these recycled polymers separately or a mixture of them? If you used a mixture of them, did you investigate results of using separate recycled polymers and reasons of differences in obtained results?
Answer: Thank you for the important question to explain the work better. The polymers indicated are found within the plastics used in this study. Consequently, all three polymers were used by inserting them within the HMA to create the PMAs. There are studies in the literature showing the compatibility of the individual polymers with the asphalt mixtures.
7-Please add some original test curves obtained from various samples analysis (creep, tensile, and so on) as in DSC analysis. You just showed final results in charts.
Answer: Thanks for this comment. We are aware about it, but authors tried to use graphs instead of charts, Nevertheless, the result was not clear due to the number of curves. Thus, it was decided to use and discuss the bar-chart reporting the analysed parameters.
8-There are numerous spelling and grammatical errors that must be corrected.
Answer: Thanks for the suggestion. The article has been revised following this observation.
Round 2
Reviewer 1 Report
The answers to the questions and the corrections in the article are satisfactory.
Author Response
Thank you very much for considering our answers satisfying.
Reviewer 2 Report
The authors do not carefully respond to the comments.
no
Reviewer 3 Report
Dear authors
Please consider the following comment. It is necessary.
1-Please add important numerical results in abstract and conclusion sections.
Abstract and conclusion must include some important numerical results.
7-Please add some original test curves obtained from various samples analysis (creep, tensile, and so on) as in DSC analysis. You just showed final results in charts.
You must show some original analysis curves for example.
Sincerely
It is OK.
Author Response
Dear authors
Please consider the following comment. It is necessary.
1-Please add important numerical results in abstract and conclusion sections.
Abstract and conclusion must include some important numerical results.
Answer: Authors reported some numerical indication referring to the proneness to accumulate permanent deformation.
7-Please add some original test curves obtained from various samples analysis (creep, tensile, and so on) as in DSC analysis. You just showed final results in charts.
You must show some original analysis curves for example.
Answer: Authors included the creep compliance curves.
Round 3
Reviewer 2 Report
The authors have replied the comments. This paper can be accepted in this state.
Author Response
Dear,
thank you very much for your feedback.
Sincerely yours
Reviewer 3 Report
Dear authors
The comment "7" was not considered completely. If you worry about prolongation of the text, you can add them in a supplementary file. Please add samples of other analysis.
Sincerely
It is Ok
Author Response
Dear,
I am really sorry about that. The authors provided the creep compliance curves, instead of bar charts. The other mechanical properties are referred to as analytical parameters [1,2,3,4,5,6]. Therefore, we reported them in bar charts drawing the error bars for each analysed material.
Sincerely yours.
- Roque, R.; Buttlar, W.G. Development of a measurement and analysis system to accurately determine asphalt concrete properties using the indirect tensile mode. Asphalt Paving Technology: Association of Asphalt Paving Technologists-Proceedings of the Technical Sessions 1992, 61, 304–332.
- Buttlar, W.G.; Roque, R. Evaluation of empirical and theoretical models to determine asphalt mixture stiffnesses at low temperatures. Association of Asphalt Paving Technologists 1996, 65, 99–141.
- Roque, R.; Zhang, Z.; Sankar, B. Determination of Crack Growth Rate Parameters Of Asphalt Mixtures Using the Superpave IDT. Journal of the Association of Asphalt Paving Technologists 1999, 68, 404–433.
- Roque, R.; Birgisson, B.; Drakos, C.; Dietrich, B. Development and field evaluation of energybased criteria for top-down cracking performance of hot mix asphalt. Journal of the Association of Asphalt Paving Technologists 2004, 72, 229–260.
- Roque, R.; Birgisson, B.; Sangpetngam, B.; Zhang, Z. Hot mix asphalt fracture mechanics: A fundamental crack growth law for asphalt mixtures. Journal of the Association of Asphalt Paving Technologist 2002, 71, 816–827.
- Zhang, Z.; Roque, R.; Birgisson, B.; Sangpetngam, B. Identification and verification of a suitable crack growth law. Journal of the Association of Asphalt Paving Technologists 2001, 70, 206–2041.